

# IQGAP3 in clear cell renal cell carcinoma contributes to drug resistance and genome stability

Wen Li[1,2,*], Zhifeng Wang[3,*], Hanlin Wang[1], Jian Zhang[4], Xiaobin Wang[1,2], Shaojun Xing[1,5] and Si Chen[1]

[1] Health Science Center, School of Medicine, Shenzhen University, Shenzhen, Guangdong, China
[2] Carson International Cancer Centre, Shenzhen University General Hospital and Shenzhen University Clinical Medical Academy Centre, Shenzhen University, Shenzhen, Guangdong, China
[3] Department of Urology, Henan Provincial People's Hospital, Zhengzhou University People's Hospital, Henan University People's Hospital, Zhengzhou, Henan, China
[4] Department of Pharmacy, Health Science Center, Shenzhen University, Shenzhen, Guangdong, China
[5] Marshall Laboratory of Biomedical Engineering, Shenzhen University, Shenzhen, Guangdong, China
[*] These authors contributed equally to this work.

## ABSTRACT

**Background**. Clear cell renal clear cell carcinoma (ccRCC) is resistant to most chemotherapeutic drugs and the molecular mechanisms have not been fully revealed. Genomic instability and the abnormal activation of bypass DNA repair pathway is the potential cause of tumor resistance to radiotherapy and chemotherapy. IQ-motif GTPase activating protein 3 (IQGAP3) regulates cell migration and intercellular adhesion. This study aims to analysis the effects of IQGAP3 expression on cell survival, genome stability and clinical prognosis in ccRCC.

**Methods**. Multiple bioinformatics analysis based on TCGA database and IHC analysis on clinical specimens were included. Quantitative real-time polymerase chain reaction (qRT-PCR) and western blot (WB) were used to determine protein expression level. MTT assay and 3D spheroid cell growth assay were used to assess cell proliferation and drug resistance in RNAi transfected ccRCC cells. Cell invasion capacity was evaluated by transwell assay. The influence of IQGAP3 on genome instability was revealed by micronuclei number and $\gamma$ H2AX recruitment test.

**Results**. The highly expressed IQGAP3 in multiple subtypes of renal cell carcinoma has a clear prognostic value. Deletion of IQGAP3 inhibits cell growth in 3D Matrigel. IQGAP3 depletion lso increases accumulated DNA damage, and improves cell sensitivity to ionizing radiation and chemotherapeutic drugs. Therefore, targeting DNA damage repair function of IQGAP3 in tumorigenesis can provide ideas for the development of new targets for early diagnosis.

Corresponding authors
Shaojun Xing,
shaojun-xing@szu.edu.cn,
shaojun.xing@yahoo.com
Si Chen, chensi@szu.edu.cn

# INTRODUCTION

According to the pathological features, renal carcinoma can be divided into clear cell renal cell carcinoma (ccRCC), papillary renal cell carcinoma (PRCC), chromophobe renal

cell carcinoma (CRCC) and a few tumors found in the kidney (collecting duct carcinoma, medullary renal cell carcinoma and urothelial carcinoma) (*Jonasch, Gao & Rathmell, 2014*). The transparent or eosinophilic cytoplasm is a typical feature in ccRCC, which accounts for about 70% of all renal cell carcinomas (*Jonasch, Walker & Rathmell, 2021*). Although early local ccRCC can be treated by partial or radical nephrectomy, ablation or regular radiation (*Ljungberg et al., 2015*; *Pierorazio et al., 2015*), up to one-third of patients develop into metastatic renal cell carcinoma which is difficult to conventional chemotherapy (*Ljungberg et al., 2015*; *Hsieh et al., 2017*).

Abnormal DNA damage response (DDR) causes genomic instability to promote tumorigenesis. DNA mismatch repair (MMR) is suppressed in ccRCC by several ways: (1) The regulation of histone deacetylase HDAC6 by VHL gene deletion and ubiquitin-proteasome dependent MSH2 degradation (*Dere et al., 2015*; *Zhang et al., 2014*); (2) the haploid dose deficiency of MLH1 caused by deletion of chromosome 3p fragment (*Wang et al., 2012*); (3) the weakened MSH6 recruitment and transcriptional coupled repair by H3K36me3 depletion, which acts as a recognition target (*Jonasch, Walker & Rathmell, 2021*; *Li et al., 2013*). Homologous recombination (HR) repair is also suppressed in ccRCC due to the loss of VHL protein or ubiquitination modification (*Metcalf et al., 2014*). Inhibition of hyperactive DDR enhances the sensitivity of tumors to chemotherapy drugs causing DNA double-strand break (DSB, such as ionizing radiation, bleomycin and cisplatin), and minimize non targeted toxicity to normal tissues (*Ferguson et al., 2015*). The therapeutic strategy is to target key repair signals and promote cell death by increasing the number of DSBs (*Pilie et al., 2019*).

The IQ-motif GTPase activating protein (IQGAP) includes IQGAP1, IQGAP2 and IQGAP3 in mammalian cells, which are closely related to intercellular adhesion, cell division, cell movement and migration, endocytosis and exocytosis (*Shannon, 2012*; *Noritake et al., 2005*; *Brown & Sacks, 2006*; *White, Erdemir & Sacks, 2012*). These isoforms share similar domains and bind the Rho family member CDC42 in a GTP-dependent manner to regulate the actin cytoskeleton (*Mosaddeghzadeh et al., 2021*; *Briggs & Sacks, 2003*). IQGAP2 and IQGAP3 have unique functions compared with IQGAP1. IQGAP2 contains all the domains of IQGAP1 with diverse interaction partners. Different from IQGAP1, IQGAP2 binds CDC42 but not RhoA or RAS (*Brill et al., 1996*). The tissue distribution and subcellular localization between the three isotypes showed significant difference. IQGAP1 was expressed in almost all tissues and mainly distributed at the cell contact sites at the cell edge, while IQGAP2 was significantly expressed in liver, stomach, platelets, prostate, kidney, thyroid, stomach, testis and salivary gland, and showed strong intranuclear localization (*Briggs & Sacks, 2003*; *Yamashiro, Noguchi & Mabuchi, 2003*; *White, Brown & Sacks, 2009*). IQGAP3 is mainly expressed in brain, testis, small intestine, lung and colon (*White, Brown & Sacks, 2009*).

The function of IQGAP3 involved in DNA damage repair has been gradually revealed. In lung cancer, IQGAP3 directly bind repair protein Rad17 to regulate its expression and localization at the DNA damage site, so as to promote DNA repair (*Zeng et al., 2020*). In cervical cancer, IQGAP3 regulates cell cycle and promotes genome stability through MMS19/XPD/CAK axis (*Leone et al., 2021*). Compared with the other two widely studied

isoforms, only IQGAP3 showed increased expression in different subtypes of renal cancer than normal tissue, indicating its general function in renal cancer. More studies are needed to elucidate the interaction partners and biological roles of IQGAP3. The expression of IQGAP3 in renal cell carcinoma, its correlation with prognosis or chemoradiotherapy sensitivity, the molecular mechanism involved in tumor malignant progression will help further biomarkers identification and combination therapy exploration.

## MATERIALS & METHODS

### Human myocardial tissue collection
Seven pairs of tumor and adjacent normal tissues were collected from the department of urology, Henan Provincial People's Hospital. The study was approved by the medical ethics committee of Henan Provincial People's Hospital (No. 2019074) following the Declaration of Helsinki. The experiments were undertaken with the understanding and written consent of each subject. The participants allowed the researchers to use their tissue during the tumor resection and conduct the study accordingly. The patient information was listed in Table 1.

### Cell culture and RNAi transfection
Human clear cell adenocarcinoma cell lines 786-O and ACHN were seeded at 37 °C and 5% $CO_2$ in RPMI-1640 and DMEM medium, respectively. All mediums were supplemented with 10% FBS and 1% Penicillin/Streptomycin. The cell lines were obtained from Shanghai Zhong Qiao Xin Zhou Biotechnology and were went through mycoplasma testing every month. The Lipofectamine RNAiMAX reagent (Invitrogen) was used to transfect siRNAs (50 nM) for 72 to 96 h. The siRNA sequences targeting IQGAP3 were as follows:
siIQGAP3-1#: 5′-CGUCCGAACUGGCCAAAUA-3′;
siIQGAP3-2#: 5′-GGGUGUGGCUGUCAUGAAA-3′.

### Antibodies
The human IQGAP3 antibody was obtained from Proteintech (25930-1-AP, 1:1000). Human GAPDH antibody was obtained from Proteintech (10494-1-AP, 1:1000). Human Integrin Alpha 6 antibody was obtained from Proteintech (27189-1-AP, 1:1000). Human Twist antibody was obtained from Proteintech (11752-1-AP, 1:1000). Human Slug antibody was obtained from Proteintech (12129-1-AP, 1:1000). Human Vimentin antibody was obtained from Proteintech (10366-1-AP, 1:1000). Human Phospho-H2AX-S139 antibody was obtained from Abclonal (AP0687, 1:1000).

### Western blotting
The collected cells were centrifuged and lysed in RIPA buffer (150 mM NaCl, 50 mM Tris–HCl (pH 7.4), 1% Triton X-100, 1% sodium deoxycholate, 0.1% SDS) with 1% PMSF for 30 min. The supernatant was separated by 13,000 g centrifugation at 4 °C for 20 min. The protein samples were denatured at 100 °C for 10 min and loaded in SDS-PAGE gel. After being transferred to a PVDF membrane, the blocking was performed with 5% skimmed milk for at room temperature 1 h. The membrane was incubated overnight at 4 °C with

**Table 1  Patient information.**

| Patient | Gender | Age | Capsule infiltration | Histological grade | TNM | TNM Stage |
|---------|--------|-----|----------------------|--------------------|-----|-----------|
| 1 | Male | 37 | No | Grade 3 | T1aN0M0 | Stage I |
| 2 | Female | 52 | Yes | Grade 2 | T3N0M0 | Stage III |
| 3 | Female | 55 | No | Grade 1 | T1aN0M0 | Stage I |
| 4 | Female | 60 | Yes | Grade 3 | T1bN0M0 | Stage I |
| 5 | Male | 50 | No | Grade 3 | T1aN0M0 | Stage I |
| 6 | Male | 48 | No | Grade 3 | T1aN0M0 | Stage I |
| 7 | Male | 48 | No | Grade 1-2 | T1aN0M0 | Stage I |

the primary antibodies and incubated at room temperature with the secondary antibody for 1 h. Signal detection was performed by enhanced chemiluminescence (PerkinElmer).

## Cell proliferation and survival assay

Cell proliferation was detected by MTT assay. Briefly, $2 \times 10^3$ cells were seeded in 96 well plate for 1 to 6 days. MTT reagent (Sigma-Aldrich) was added at the concentration of 5 mg/ml, followed by 4 h incubation at 37 °C. The culture supernatant was discarded and 100 µl DMSO (Sigma-Aldrich) was added. After 10 min of oscillation, 490 nm wavelength was selected on the enzyme-linked immunosorbent monitor to measure the light absorption value.

For the ionizing radiation sensitivity test, 200 to 5000 cells were seeded in six well plate, followed by several doses of X-ray irradiation. The number of clones was counted after 14 days of cell culture. For drug sensitivity test, 2000 cells were seeded in 96 well plate with several dosed of cisplatin, camptothecin and doxorubicin (Selleck). The surviving cells were measured by MTT method after 48 h of culture.

## 3D spheroid cell growth assay

For the 3D spheroid cell growth assay, $1 \times 10^3$ cells were seeded in 24 well plates with ultra-low protein adsorption. The cell images were taken and recorded by light microscope after 6 days, 12 days and 14 days.

## Immunohistochemistry (IHC)

IHC staining was performed on renal cell carcinoma tissue. Tissue sections were dewaxed in xylene and rehydrated in graded ethanol (100%, 95%, 80% and 70% ethanol for 10 min). The antigen was recovered by heat induced epitope recovery method. The slices were treated by 10 mmol/l EDTA (pH 8.0) at 98 °C for 15 min. 3% hydrogen peroxide were used in methanol at 37 °C for 15 min to quench the endogenous peroxidase activity, followed by blocking with 5% bovine serum albumin. The incubation condition of primary antibody was 4 °C overnight.

## Immunofluorescence

The growing cells were inoculated on the sterilized coverslips and cultured at 37 °C and 5% $CO_2$ for 24 h. The medium was removed when the cell grew to about 70% and washed

with PBS. The cells were fixed at room temperature with 4% paraformaldehyde for 15 min. After washing with PBS for three times, Triton was added at the final concentration of 0.3% to treat the cell at room temperature for 20 min. After blocking in 5% BSA for 30 min, the cells were stained with primary antibodies (diluted in 1% BSA) at room temperature for 2 h. Cells were washed with PBST (PBS with 0.1% Tween-20) three times and incubated with fluorescence-conjugated secondary antibodies at room temperature for 1 h. After being washed three times with PBST, cells were mounted with antifade mounting medium with 2-(4-Amidinophenyl)-6-indolecarbamidine dihydrochloride (DAPI). The slides were observed by confocal microscope, and the fluorescence intensity were calculated by ImageJ software.

## TCGA database analysis

The sequencing results (HTSeq FPKM data) were obtained and analyzed from TCGA database (https://portal.gdc.cancer.gov/). IQGAP3 expression between normal and tumor tissues was analyzed by UALCAN database (http://ualcan.path.uab.edu). Survival analysis between patients with low and high IQGAP3 expression was performed by Kaplan Meier database (https://kmplot.com/analysis/). Timer database (http://timer.cistrome.org/) was used to explore the relationship between IQGAP3 expression and immune infiltration. Student $t$-test and log rank test were used for data statistics.

## Transcriptome sequencing and analysis

Total RNA isolation by Trizol, mRNA enrichment with oligo DT magnetic beads, mRNA fragmentation and cDNA synthesis were all processed according to the manufacturer's protocol. The cDNA was complemented and repaired. After amplification, the RNA library was sequenced by Illumina pe150 in Shenzhen Haplox company. The reference genome used was GRCh37 (hg19). The FPKM (Fragments Per Kilobase of exon model per Million mapped fragments) of each gene was calculated according to gene length. Differential expression analysis was performed using the DESeq R package (*Yu et al., 2012*).

## Statistical analysis

Each experiment was validated by three independent replicates. Unpaired two tailed Student $t$-test was used to analyze the statistical significance. The experimental values are expressed as the mean $\pm$ standard deviation (SD). Statistical significance was analyzed by GraphPad Prism 6.0 software (ns, $P > 0.05$; *, $P < 0.05$; **, $P < 0.01$; ***, $P < 0.001$).

# RESULTS

## IQGAP3 was highly expressed in most cancer types compared with normal tissues

The expression analysis of TCGA database showed that there was no significant difference in the expression of IQGAP1 in the cancer and adjacent tissues of the three renal cancer subtypes (data were not shown). IQGAP2 was downregulated in tumor tissues of kidney renal clear cell carcinoma (KIRC) and kidney renal papillary cell carcinoma (KIRP), and upregulated in tumor tissues of kidney chromophobe (KICH) (data were not shown). The mRNA sequencing data of IQGAP3 from 730 adjacent normal tissues and 10,363 tumor

tissues in TCGA pan cancer database were extracted. Comparing unpaired samples and paired samples, IQGAP3 was highly expressed in most tumor types (Figs. 1A, 1B). All the three subtypes of renal cell carcinoma showed significantly enhanced IQGAP3 expression in tumor tissues than that in normal tissues (Fig. 1C). Due to the small number of samples contained in KICH, the followed bioinformatics analysis mainly focused on the common subtypes KIRC and KIRP. Gene expression profiles were obtained by high-throughput gene array analysis in GEO database (https://www.ncbi.nlm.nih.gov/geoprofiles/). The expression of IQGAP3 in 27 pair (Series: GSE66272) and 72 pair (Series: GSE53757) of ccRCC tumor tissues and matched normal tissues at different disease stages were extracted and analyzed. IQGAP3 increased significantly in tumor tissues at different stages (Figs. 1D, 1E). Interestingly, IQGAP3 was highly expressed in both cancer and adjacent paired samples with (13 pairs of samples, Series: GSE66271, Fig. 1F) or without metastasis (14 pairs of samples, Series: GSE66270, Fig. 1G). In patients with metastatic ccRCC, the expression is higher in tumor tissue than normal tissue. Immunohistochemistry also showed that the expression of IQGAP3 was high in tumor tissues, which is consistent with the analysis of the database (Fig. 1H).

## Correlation between IQGAP3 expression and clinical features

According to clinical features in TCGA database, the higher the TNM (tumor-node-metastasis) stage, histological grade and pathological stage, the higher IQGAP3 expression in KIRC and KIRP (Figs. 2A, 2B). 265 ccRCC tumor samples from GEO database were compared (Series: GSE73731). The expression of IQGAP3 was higher in high-grade samples (Fig. 2C). The diagnostic value of IQGAP3 mRNA level was evaluated by ROC (receiver operating characteristic) curve and the area under the ROC curve (AUC). The AUC value of IQGAP3 were 0.934 and 0.939 in KIRC (Fig. 2D) and KIRP (Fig. 2E) respectively, which showed high diagnostic value. The mRNA expression of IQGAP3 have similar diagnostic value in different stages and grades.

## Correction between IQGAP3 expression with prognosis and immune cell infiltration in two subtypes of renal cell carcinoma

Kaplan–Meier analysis showed that the high expression of IQGAP3 was correlated with low OS (Overall Survival), DSS (Disease Specific Survival) and PFS (Progression Free Survival) both in KIRC and KIRP (Figs. 3A, 3B). The "immune gene" module of Timer database was used to explore the relationship between IQGAP3 expression and immune infiltration. In KIRC, IQGAP3 expression was positively correlated with infiltrated Th2 cells, Treg cells, NK (CD56+) cells, Th1 cells, aDC cells, T cells, macrophages and B cells, but negatively correlated with iDC cells, NK cells, Tgd cells, pDC cells, Th17 cells and mast cells (Fig. 3C). In KIRP, IQGAP3 expression was positively correlated with infiltrated Th2 cells, aDC cells, pDC cells and T helper cells, as well as a negative correction with DC cells, cytotoxic cells, neutrophils, Tem cells, CD8 T cells, mast cells, iDC cells, eosinophils and macrophages (Fig. 3D).

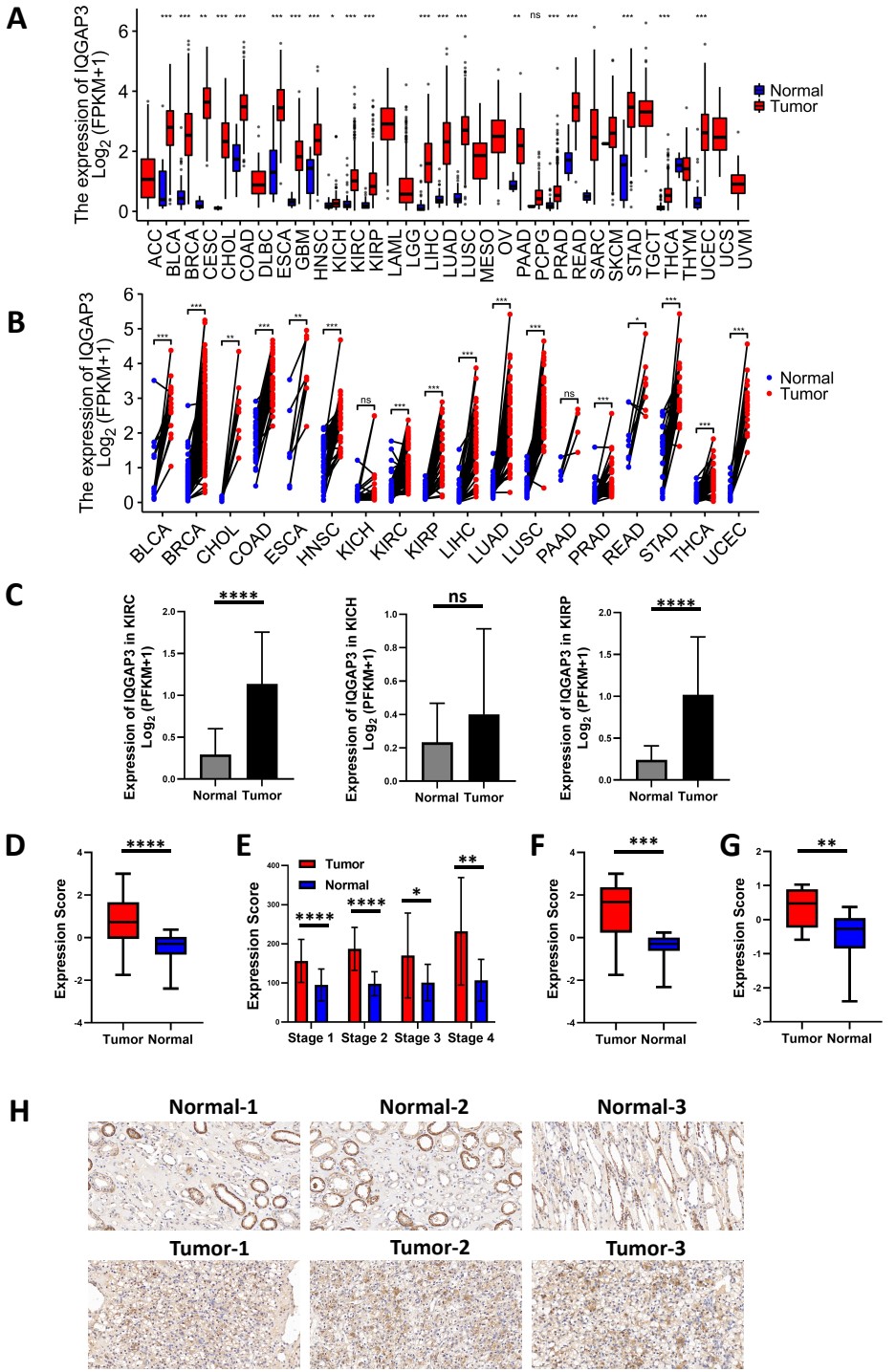

**Figure 1  IQGAP3 expression was increased in cancer than normal tissues.** (A–B) The unpaired (A) and paired (B) mRNA sequencing data of IQGAP3 in tumor and adjacent normal tissues in TCGA pan cancer database. (C) Expression of IQGAP3 in TCGA (KIRC, KICH and KIRP) 

**Figure 1 (…continued)**
tumor and normal samples. (D) IQGAP3 expression in tumor and normal samples from GSE66272 dataset. (E) IQGAP3 expression in tumor and normal samples from GSE53757 dataset. (F) IQGAP3 expression in tumor and normal samples from GSE66271 dataset. (G) IQGAP3 expression in tumor and normal samples from GSE66270 dataset. (H) Immunohistochemical staining of IQGAP3 expression in ccRCC patient tissues.

## The deletion of IQGAP3 in ccRCC inhibit cell proliferation

IQGAP3 depletion was performed by siRNA transfection in ccRCC cell lines 786-O and ACHN (Fig. 4A). Cell proliferation was reduced after IQGAP3 depletion in both cell lines (Fig. 4B). Cell clone formation and cell metastasis were not affected (data were not shown). The cells present cell agglomerates in the 3D cell culture dish (Fig. 4C). At the early stage of culture (day 6), IQGAP3 knockdown had no significant effect on the growth of cell spheres. At the late stage of culture (day 12), deletion of IQGAP3 significantly inhibited the growth of 3D cell spheres (Fig. 4D). Cell growth in 3D culture was significantly suppressed after IQGAP3 depletion after 14 days both in 786-O and ACHN cells (Fig. 4E).

## The depletion of IQGAP3 increased genomic instability

In the process of cell mitosis, some chromosome breaks will produce centromere-free chromosome fragments, which are wrapped in the nuclear membrane to form a micronucleus (MN) structure with a diameter less than 1/3 of the normal nucleus. In order to determine the effect of IQGAP3 on genomic stability in ccRCC, the formation of MNs was counted after 4 Gy ionizing radiation (IR) and recovery for 4 h (Fig. 5A). After IQGAP3 knockdown, the number of MNs in cells increased regardless of IR treatment (Fig. 5B). Phosphorylation of serine (Ser) at position 139 of histone H2AX ($\gamma$ H2AX) is considered to be a marker of DNA breakage (Fig. 5C). As the initial signal molecule of damage induction, $\gamma$ H2AX recruits a series of DNA damage repair proteins at the damage site to start the DNA repair cascade. After IR irradiation, the intensity and quantity of $\gamma$H2AX were increased in 786-O and ACHN cells (Figs. 5D, 5E). With the increase of recovery time after radiation, DNA damage in IQGAP3 knockdown cells accumulated continuously (Fig. 5F). These results show that IQGAP3 can promote genome stability.

## The depletion of IQGAP3 increases the sensitivity of ccRCC cells to radiation and chemotherapy drugs

The cells were treated with different doses of IR, and the cell survival was counted. After IQGAP3 knockdown, the cell survival rate decreased significantly and the cell radiosensitivity increased (Fig. 6A). The three common chemotherapeutic drugs, cisplatin, camptothecin and doxorubicin were selected to test IC50 value. The sensitivity of cells to these drugs increased after IQGAP3 knockdown (Fig. 6B). The results show that IQGAP3 promotes genome stability, which may be the reason for chemoradiotherapy resistance of ccRCC.

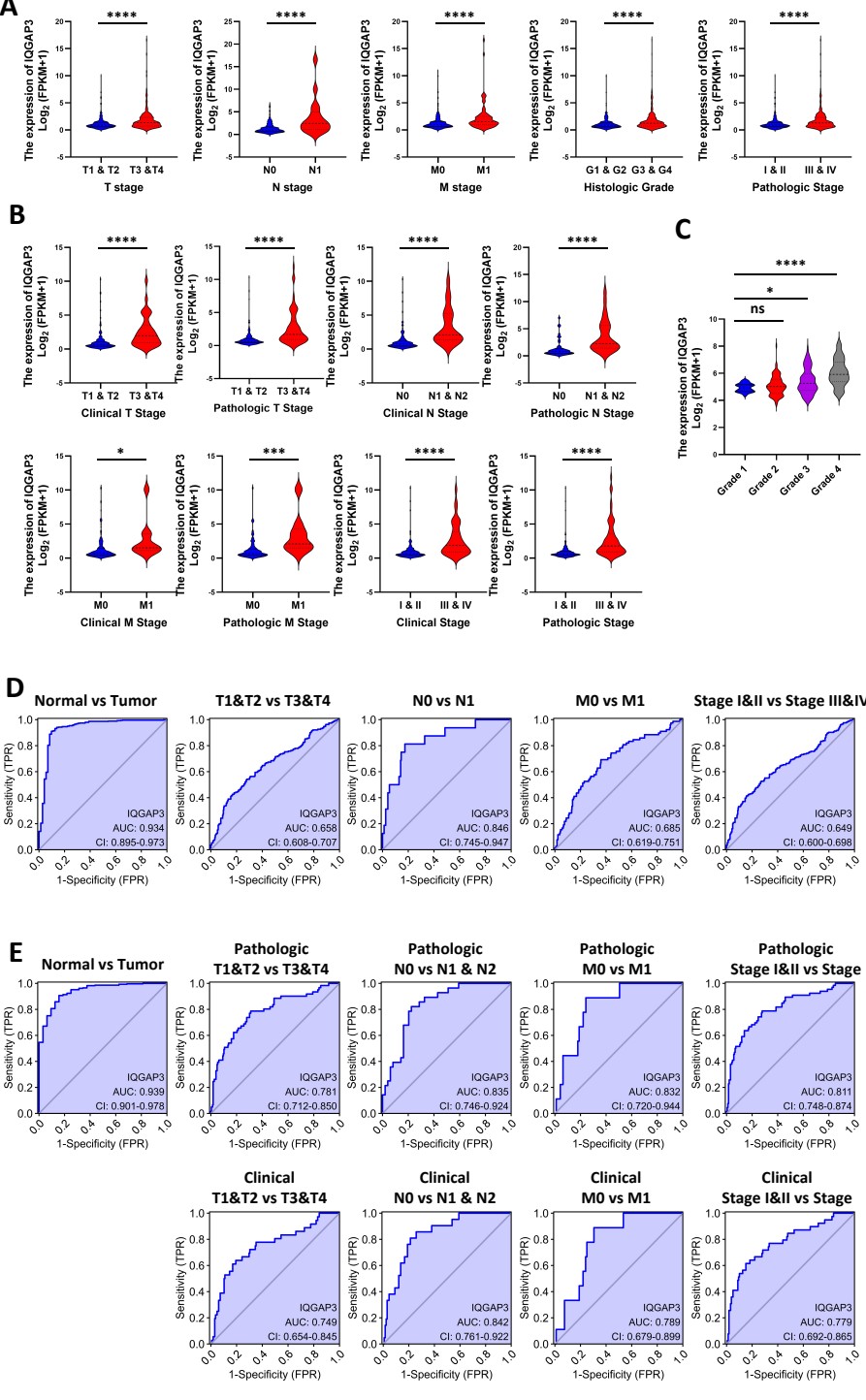

**Figure 2** **Correlation between IQGAP3 expression and clinical features.** (A) IQGAP3 expression in different TNM stages, histological grades and pathological stages of KIRC. (B) IQGAP3 expression in different IQGAP3 expression in different clinical TNM stages, pathologic (continued on next page...)
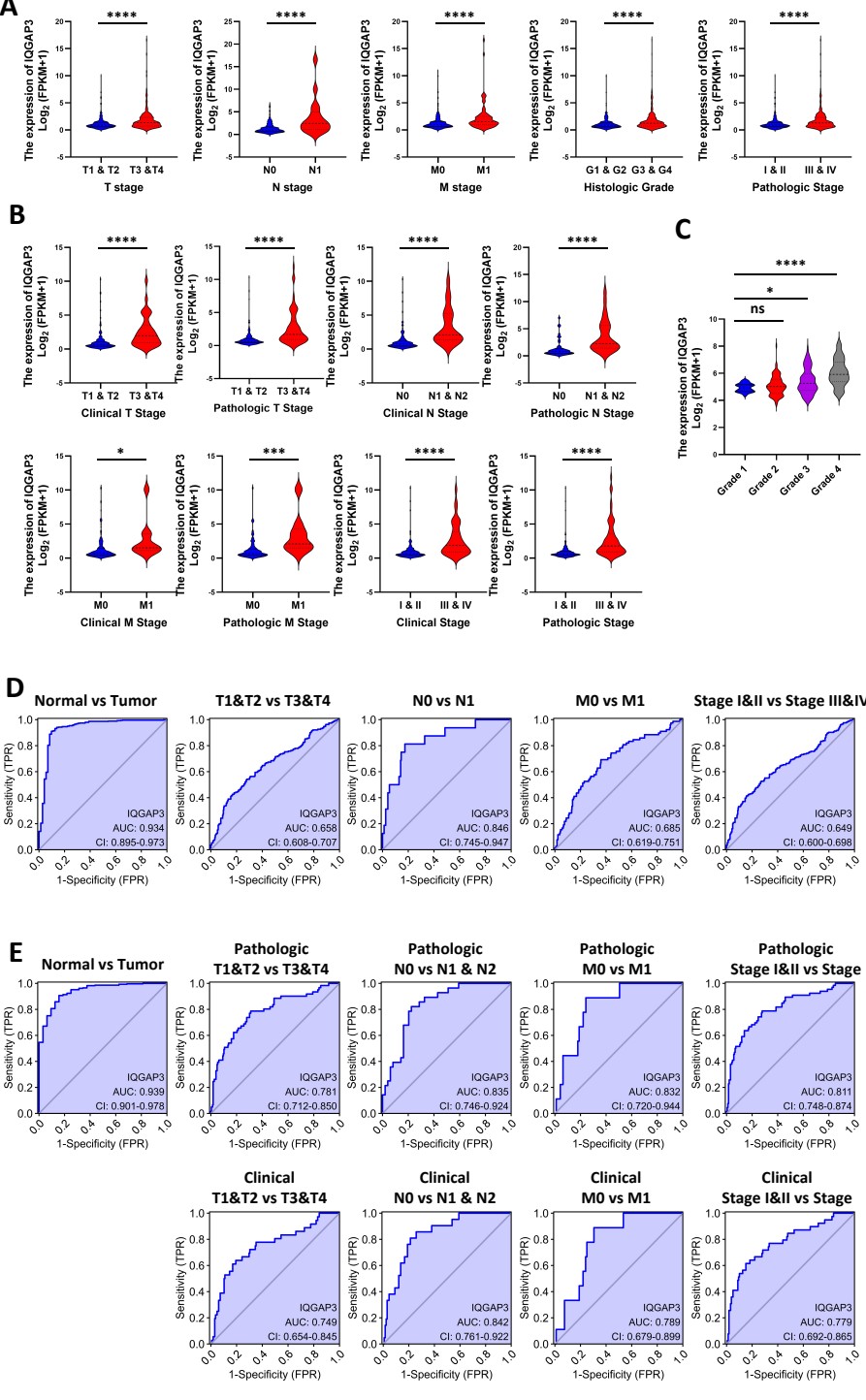

**Figure 2 (…continued)**
TNM stages, all clinical stages and all pathological stages of KIRP. (C) The expression of IQGAP3 in different grades from GSE73731 dataset. (D) ROC curve of IQGAP3 expression in different TNM stages and histological grades of KIRC. (E) ROC curve of IQGAP3 expression in clinical TNM stages, pathologic TNM stages, all clinical stages and all pathological stages of KIRP.

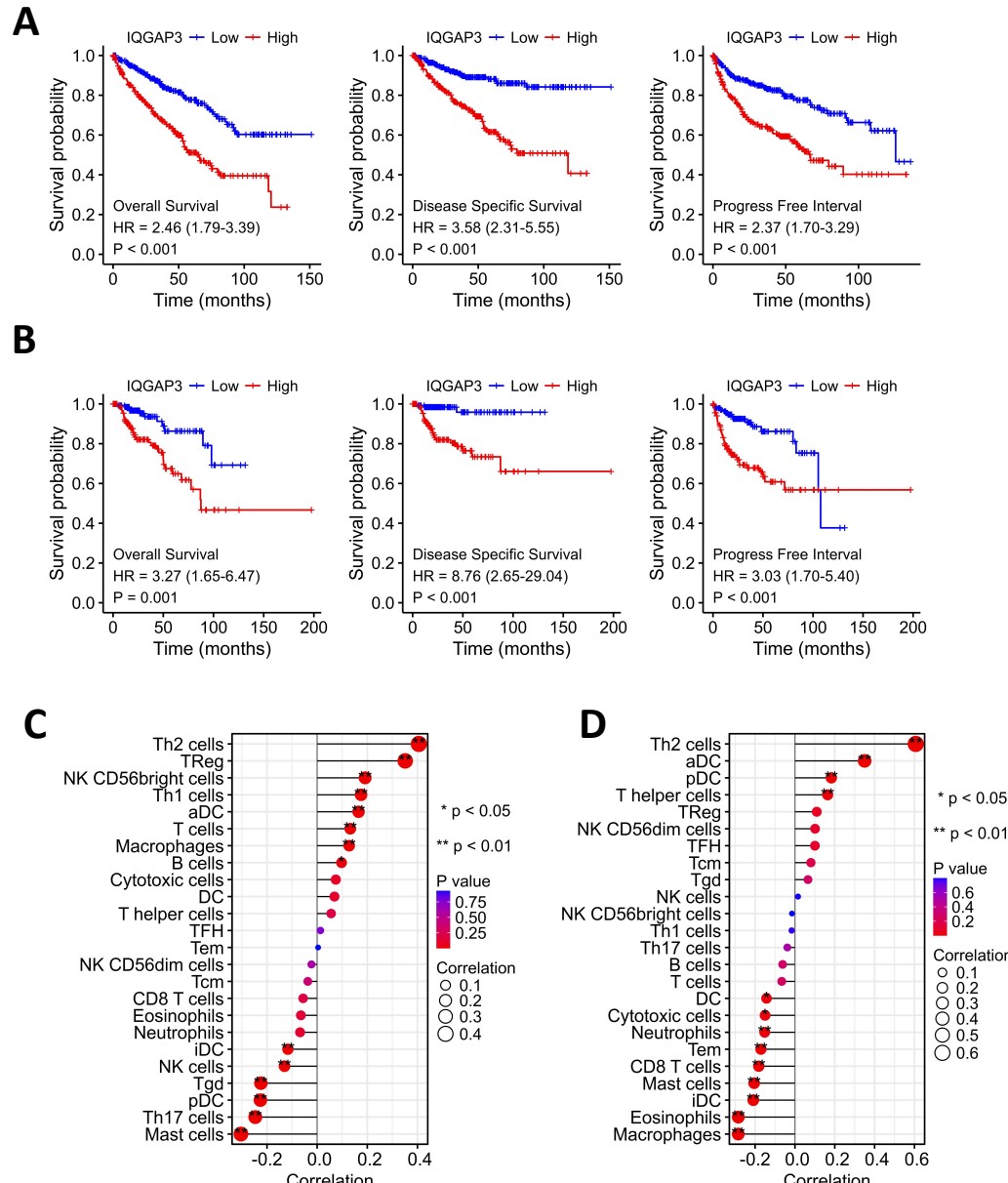

**Figure 3** **Correction between IQGAP3 expression with prognosis and immune cell infiltration.** (A) The Kaplan–Meier plot with OS, DSS and PFS in KIRC. (B) The Kaplan–Meier plot with OS, DSS and PFS in KIRP. (C) The correlation between IQGAP3 expression and infiltrated immune cells in KIRC. (D) The correlation between IQGAP3 expression and infiltrated immune cells in KIRP.

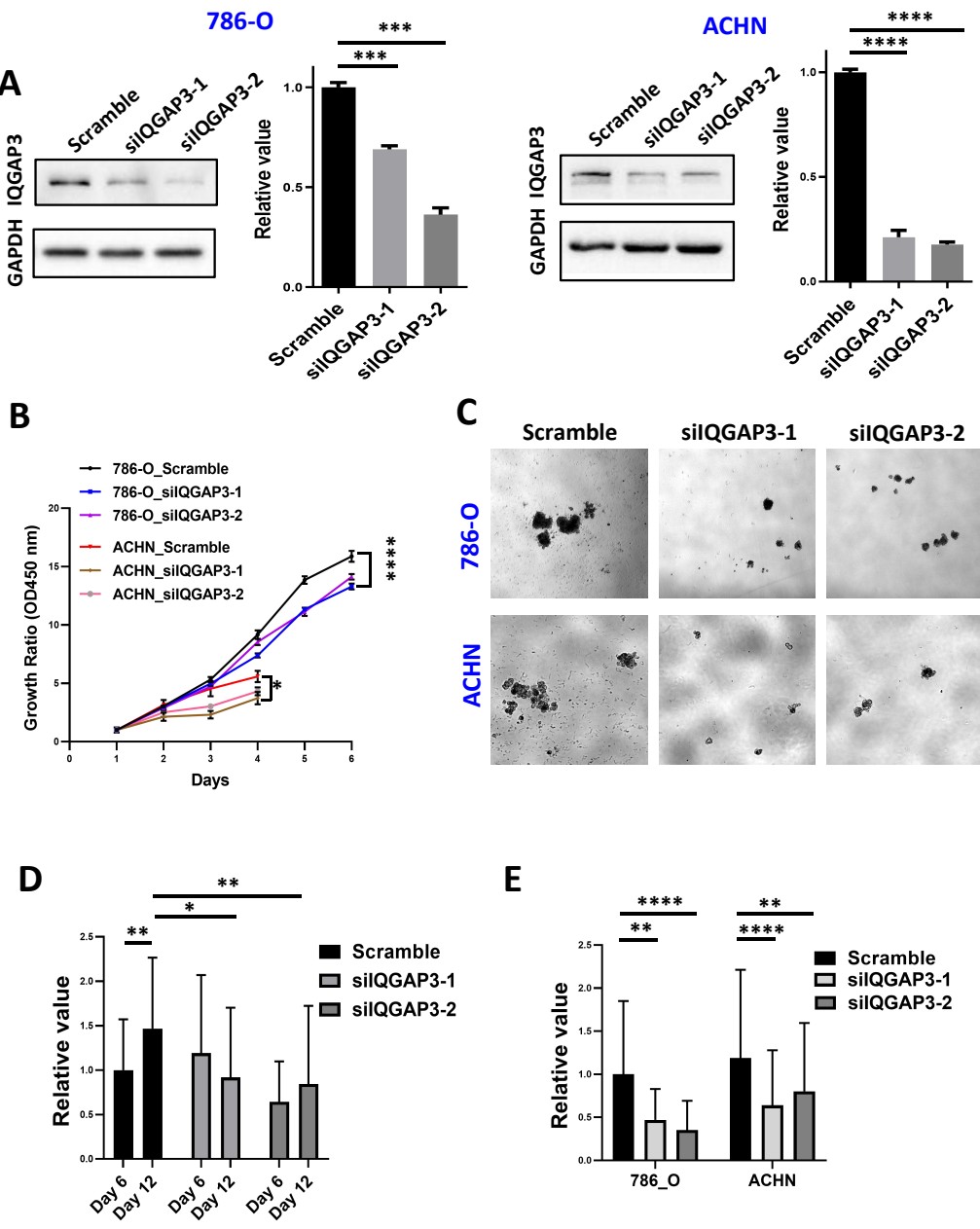

**Figure 4 Depletion of IQGAP3 inhibit cell proliferation.** (A) Effect of siRNA transfection on the expression of IQGAP3. (B) Cell proliferation after IQGAP3 depletion in 786-O and ACHN cells. (C) Schematic diagram of 3D cultured cell agglomerates. (D) The relative values of the cell sphere volume after 6 and 12 days of culture in the ultra-low adhesion cell culture dish. (E) The relative values of the cell sphere volume after 14 days of culture in the ultra-low adhesion cell culture dish in 786-O and ACHN cells.

## Enrichment of migration and drug metabolism related genes after IQGAP3 depletion

In order to study the signal pathways of IQGAP3 stabilizing genome and then participating in the inhibition of 3D growth in ccRCC, the IQGAP3 knockdown cells were sequenced

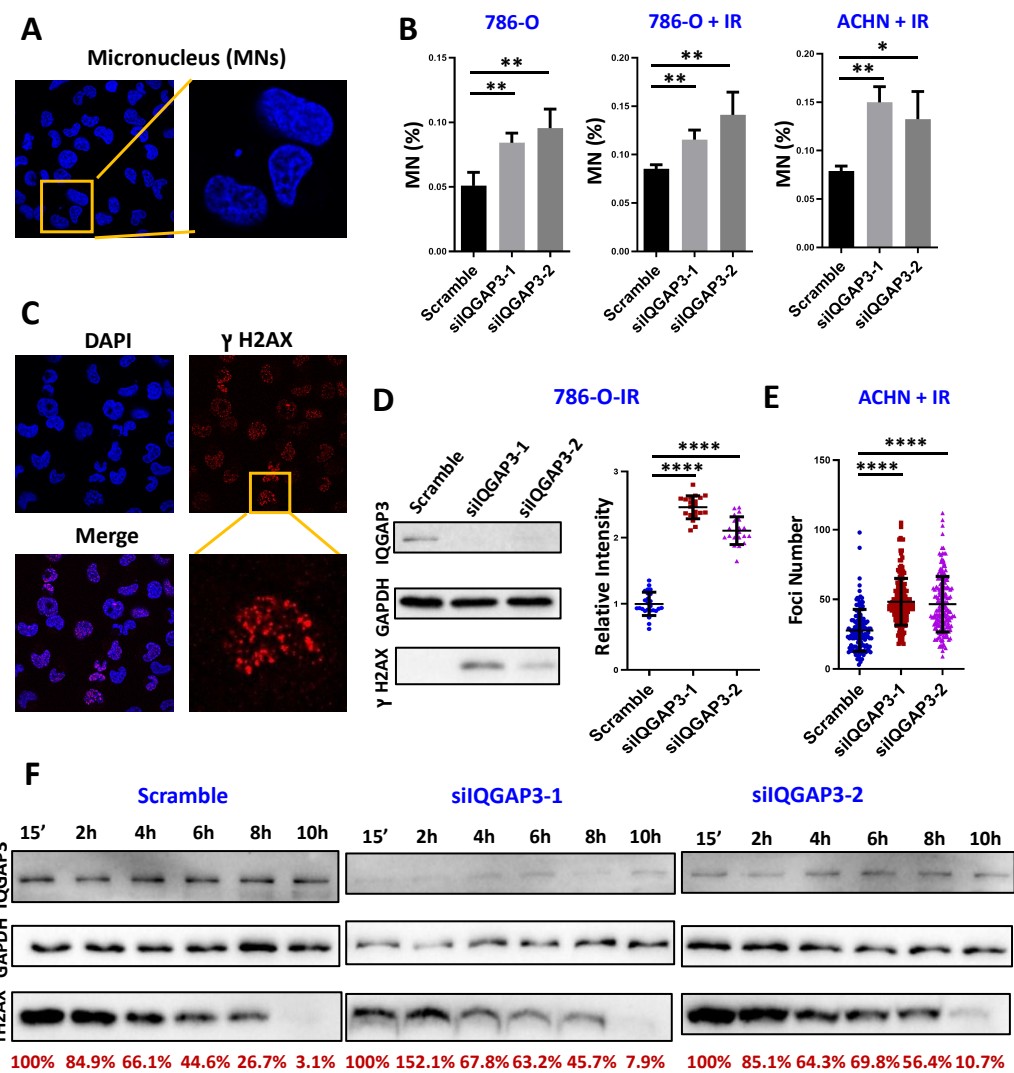

**Figure 5** **IQGAP3 promoted genomic stability.** (A) Intracellular micronucleus (MN) structure after IR irradiation. (B) Statistics of the number of micronuclei in untreated and IR irradiated cells. (C) Immunofluorescence assay to test γ H2AX recruitment at DNA damage sites. (D) After IR treatment, the intensity of γ H2AX in 786-O cells. (E) γ H2AX foci number in ACHN cells. (F) The γ H2AX intensity at different recovery times after radiation. The remaining γ H2AX content is the relative value to GAPDH.

to compare the global gene expression profile. In total of 462 differentially expressed genes under the condition of fold change ≥ 2 and $p < 0.05$ were found, including 278 upregulated genes and 184 downregulated genes. The Kyoto Encyclopedia of Genes and Genomes (KEGG) was used for gene function research of these differentially expressed genes. The bubble chart displayed the affected genes were enriched in extracellular matrix (ECM)-receptor interaction, drug metabolism, regulation of actin cytoskeleton and cell adhesin molecules after IQGAP3 depletion (Fig. 6C). These results provide potential mechanistic insights into the promotion role of IQGAP3 in cell 3D growth and drug

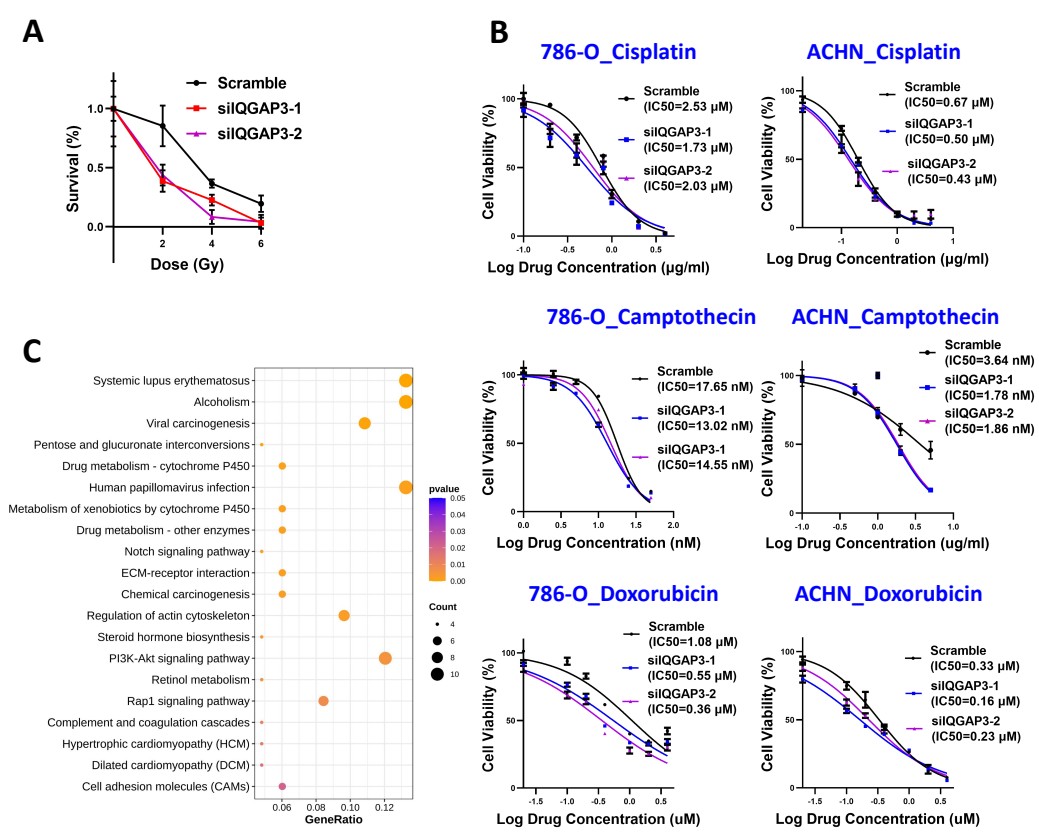

**Figure 6** **Effect of IQGAP3 on drug sensitivity.** (A) Cell survival rate under different doses of ionizing radiation. (B) Cell survival rate and IC50 value of cisplatin, camptothecin and doxorubicin. (C) Dot plot ranking for the significant enrichment pathways according to KEGG analysis between control and IQGAP3 depletion groups.

resistance in ccRCC. However, the interacting proteins and specific regulatory mechanisms need to be further studied.

## DISCUSSION

IQGAP scaffold protein is evolutionarily conservative in eukaryotes and contributes to the regulation of cytoskeleton, intracellular signal transduction and intercellular interaction. IQGAPs are usually used as scaffold proteins, which is related to different cytoskeleton content. Single molecule imaging has confirmed that the combination of IQGAP1 and its binding proteins with actin can promote cell migration and adhesion (*Hoeprich et al., 2022*). Although the three proteins of IQGAP family have similar structure and high sequence homology, the difference of their binding proteins lead to the regulation difference of downstream signal in normal or disease state. The interaction of RHO-GTPase with IQGAPs is selective. IQGAP1 and IQGAP2 bind CDC42 and RAC1, but not RIF, RHOD or RHO-like proteins (*Mosaddeghzadeh et al., 2021*). Through the binding to E-cadherin and β-catenin, IQGAP1 reduces the interaction between cadherin system and cytoskeleton

to weaken the cell–cell attachment (*Kuroda et al., 1998*). Ca$^{2+}$ enhances the affinity of calmodulin and IQGAP1, reducing the transcriptional activity of $\beta$-catenin and E-cadherin dependent adhesion (*Briggs, Li & Sacks, 2002*; *Li et al., 1999*). Based on the above targets, IQGAP1 was found to promote invasion in tumors by attenuating E-cadherin-dependent cell adhesion (*Li et al., 1999*). The core factors Raf, MEK and ERK1/2 in the mitogen activated protein kinase (MAPK) pathway promote phosphorylation dependent signaling cascades by directly binding IQGAP1 (*Ren, Li & Sacks, 2007*). Disruption the interaction of IQGAP1 with ERK1/2 inhibits Ra s and RAF driven tumorigenesis (*Jameson et al., 2013*). IQGAP1 enhances the nuclear localization of $\beta$-catenin through its interaction with pathway proteins, thereby mediating the activation of cytoplasmic Wnt signaling (*Goto et al., 2013*). IQGAP1 is involved in the construction of the whole PI (3) k-Akt pathway, and the blocking of its interaction with PI (3) K inhibit tumor cell survival (*Choi et al., 2016*). Therefore, IQGAP1 plays an important role in cancer development, and anti-tumor therapy targeting IQGAP1 interacting proteins or related pathways may be beneficial for tumor therapy. IQGAPs anchored on the lipid membrane stabilize a single actin filament in a curved shape, helping to form a highly curved complete actin ring (*Palani et al., 2021*). These fine structural and biophysical calculations seem to indicate that the regulation of the actin cytoskeleton by IQGAP protein family takes place in the cytoplasm, and its molecular mechanism in the nucleus remains to be explored.

The high expression of IQGAP3 promotes malignant processes such as tumor growth and invasion with different downstream signal pathways in many types of tumors, and recent studies have found that it is related to the treatment outcome. Unlike the oncogene IQGAP1, IQGAP2 is considered to be a tumor suppressor (*Smith, Hedman & Sacks, 2015*). The disruption of IQGAP2 in mice promoted the occurrence of hepatocellular carcinoma and was reversed by the deletion of both IQGAP1 and IQGAP2, indicating the opposite biological effects of the two isoforms (*Schmidt et al., 2008*). Decreased expression of IQGAP2 in prostate cancer promotes cell proliferation by activating Akt (*Xie et al., 2012*). Loss of IQGAP2 expression in gastric cancer promotes invasion and is associated with promoter methylation (*Jin et al., 2008*). The possible role of IQGAP3 in tumors is related to tumor types, and the mechanism research is still in the initial stage. IQGAP3 has been shown to be upregulated in breast cancer, pancreatic cancer, gastric cancer, hepatocellular carcinoma, colorectal cancer and bladder cancer, and is closely related to clinicopathological features, suggesting that it may be involved in tumor development (*Hua et al., 2020*; *Xu et al., 2016*; *Shi et al., 2017*; *Huang et al., 2021*; *Cao et al., 2019*; *Xu et al., 2019*). Several studies have found that IQGAP3 promotes cell growth and proliferation, cytoskeleton remodeling, cell migration and adhesion (*Huang et al., 2021*; *Jinawath et al., 2020*; *Liu et al., 2020*; *Lin et al., 2019*; *Nojima et al., 2008*). The expression level of IQGAP3 in radiation resistant breast cancer was higher than that in radiosensitivity group, which may be related to DNA repair and PI3K-Akt-mTOR signal pathway (*Hua et al., 2020*). In lung cancer, the interaction of IQGAP3 with DNA repair protein Rad17 was essential for Rad17 expression and foci formation, the Mre11-Nbs1-Rad50 complex formation, and ATM/Chk2 and ATR/Chk1 pathways activation (*Zeng et al., 2020*). IQGAP3 was also found to modulate cell cycle progression and genome stability through the interaction with

MMS19 and regulation of MMS19/XPD/CAK axis (*Leone et al., 2021*).As the latest studied protein in family members, the unique function of IQGAP3 in different tumors remains to be verified. In our study, the deletion of IQGAP3 can significantly increase the genomic instability and improve the sensitivity of cells to radiation and chemical drugs. Therefore, looking for hyperactive DNA damage repair pathways and participating proteins is a new idea to further elaborate the special metabolic reprogramming of renal cell carcinoma cells and their resistance to traditional radiotherapy and chemotherapy.

## CONCLUSIONS

In this work, IQGAP3 was overexpressed not only in many tumor types, but also in the three common subtypes of renal cell carcinoma. The higher the expression of IQGAP3 in patients with TNM or later clinical stage, and the higher the protein expression has a strong positive correlation with the poor survival rate. This suggests that IQGAP3 has good prognostic value in renal cell carcinoma and inhibitors of IQGAP3 function may prevent tumor invasion, proliferation and migration. It is a potential new biomarker and therapeutic target.

IQGAP3 can not only regulate tumor 3D growth, but also cause drug resistance by stabilizing the genome and reducing the accumulation of DNA damage during radiotherapy and chemotherapy. The further excavation of the function of IQGAP3 in DNA damage repair is the embodiment of the application of the concept of synthetic lethality in tumor treatment, which will help to guide the clinical practice of precise individual treatment.

### Funding

This work was supported by the National Natural Science Foundation of China (Grant No. 82003114 to W. Li and No. 81973531 to Jian Zhang), the fundamental research project of the Shenzhen Science and Technology Innovation Commission (Grant No. 20200812211704001 to Si Chen), the Medical Scientific Research Foundation of Guangdong Province (Grant No. A2019502 to Si Chen), the SZU Top Ranking Project (Grant No. 86000000210 to Jian Zhang and Si Chen), and grants of the Engineering Laboratory of Shenzhen Natural Small Molecule Innovative Drugs (Grant No. Shenfagai 2013180 to Jian Zhang) and the Shenzhen Science and Technology Program (KQTD20190929172538530 to Shaojun Xing). The funders had no role in study design, data collection and analysis, decision to publish, or preparation of the manuscript.

### Grant Disclosures

The following grant information was disclosed by the authors:
Natural Science Foundation of China: 82003114, 81973531.
Shenzhen Science and Technology Innovation Commission: 20200812211704001.
Medical Scientific Research Foundation of Guangdong Province: A2019502.
SZU Top Ranking Project: 86000000210.

Engineering Laboratory of Shenzhen Natural Small Molecule Innovative Drugs: 2013180. Shenzhen Science and Technology Program: KQTD20190929172538530.

## Competing Interests

The authors declare there are no competing interests.

## Author Contributions

- Wen Li conceived and designed the experiments, performed the experiments, analyzed the data, prepared figures and/or tables, authored or reviewed drafts of the article, and approved the final draft.
- Zhifeng Wang conceived and designed the experiments, performed the experiments, prepared figures and/or tables, and approved the final draft.
- Hanlin Wang performed the experiments, prepared figures and/or tables, and approved the final draft.
- Jian Zhang analyzed the data, authored or reviewed drafts of the article, and approved the final draft.
- Xiaobin Wang analyzed the data, prepared figures and/or tables, and approved the final draft.
- Shaojun Xing conceived and designed the experiments, authored or reviewed drafts of the article, and approved the final draft.
- Si Chen conceived and designed the experiments, authored or reviewed drafts of the article, and approved the final draft.

## Human Ethics

The following information was supplied relating to ethical approvals (i.e., approving body and any reference numbers):

The study was carried out following the Declaration of Helsinki and was approved by the Medical Ethics Committee of Henan Provincial People's Hospital (No. 2019074).

## Data Availability

The raw data is available in the Supplementary Files.

## Supplemental Information

Supplemental information for this article can be found online at http://dx.doi.org/10.7717/peerj.14201#supplemental-information.

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
