# Peer review of "IQGAP3 in clear cell renal cell carcinoma contributes to drug resistance and genome stability"

_PeerJ, doi:10.7717/peerj.14201_

## Round 0.1 · original submission · Minor Revisions

When assessing your paper, the reviewers identified that the study is well designed. However, additional details are needed to fully support the results and conclusions, and for the manuscript to be suitable for publication in this journal.

Reviewer 1 ·

Basic reporting

The authors have presented a well written manuscript. However, it needs to be thoroughly revised for typos throughout the main manuscript and the figures. For e.g., Line 93 states myocardial tissue. Is this a typo, or have the authors collected heart tissue?
The introduction gives sufficient background about DNA damage repair and the role of IQGAP3 in cancer. The authors can elaborate further on the role of other isoforms in cancer and why they specifically chose to study IQGAP3?
Figure quality can be improved, especially the axes labels are too small. Images from 3D culture (Figure 4C and D) and immunofluorescence (Figure 5A) are not very clear. Figure legends need to be more detailed.
Authors have appropriately supplied the raw data.

Experimental design

Authors have explored the role of IQGAP3 in chemoresistance and survival in renal cancer patients. It is however, unclear why authors chose to study only IQGAP3 isoform and not others?
It is also not clear from the writing how many times the assays were repeated for biological replicates and statistical significance.

Validity of the findings

Authors have provided robust bioinformatics data that show correlation of IQGAP3 with clinical features, prognosis and immune infiltration in renal cell carcinoma. However, the in vitro work does not provide sufficient data to provide a mechanistic basis for the role of IQGAP3. Addition of in vivo work will provide more significance to these studies.

3D assays shown in Figure 4 are inconclusive. Even the scramble controls look like they are not growing well. Some proliferation assay or viability assay must be done to claim that there is a difference in growth. Also, it will be important to see how the 3D structures look at an earlier time point of the culture. Lack of in vivo data may be supplemented by using appropriate 3D assays to test chemosensitivity of knockdown cells to the drugs. Viability assays are very important to claim there is a difference in growth/proliferation.
Are the western blots in Figure 5F mislabeled? This is incomplete information.

In Figure 6B, IC50 for Cisplatin and Camptothecin for all 3 cell lines is about the same. It does not seem that the knockdown of IQGAP3 significantly changes the sensitivity to these drugs.

Overall, the bioinformatics analysis supports the authors' claim, however, the in vitro data lacks scientific robustness to claim that IQGAP3 can regulated chemoresistance.

Reviewer 2 ·

Basic reporting

1. The article titled "High expression of IQGAP3 in clear cell renal cell carcinoma contributes to drug resistance by stabilizing the genome" is a well written manuscript. Clear and unambiguous, professional English was used throughout the text.

2. Literature references are fine.

3. Figures and tables are very good.

4. The Introduction part is fine. But I have some suggestions for the authors. Please add more background information on IQGAP3. I find it little inadequate. The authors should add few sentences to clarify more about the significance of their work.

Experimental design

1. The authors structured their research very well. Research question is relevant and meaningful.

2. Authors done a very good and detailed job in the "Methods and Materials" part.

3. "Result" section is well written and very detail oriented.

4. Though, I have some suggestions for "Discussion" part. This part is poorly written. Authors concentrated more on the background. Authors should describe properly the significance of their findings here. Please re-write this part.

Validity of the findings

No comment

---

## Round 0.2 · accepted · Accept

Thank you for submitting the manuscript.

Reviewer 1 ·

Basic reporting

Authors have addressed my comments satisfactorily.

Experimental design

Authors have addressed my comments satisfactorily.

Validity of the findings

Authors have addressed my comments satisfactorily.